# Advancing Food Preservation: Sustainable Green-AgNPs Bionanocomposites in Paper-Starch Flexible Packaging for Prolonged Shelf Life

**DOI:** 10.3390/polym16070941

**Published:** 2024-03-29

**Authors:** Federico Trotta, Sidonio Da Silva, Alessio Massironi, Seyedeh Fatemeh Mirpoor, Stella Lignou, Sameer Khalil Ghawi, Dimitris Charalampopoulos

**Affiliations:** 1Metalchemy Limited, 71-75 Shelton Street, London WC2H 9JQ, UK; sids@metalchemy.tech (S.D.S.); am@metalchemy.tech (A.M.); 2Department of Food and Nutritional Sciences, University of Reading, P.O. Box 226, Whiteknights, Reading RG6 6AP, UK; s.mirpoor@reading.ac.uk (S.F.M.); s.lignou@reading.ac.uk (S.L.); s.khalilghawi@reading.ac.uk (S.K.G.); d.charalampopoulos@reading.ac.uk (D.C.)

**Keywords:** bionanocomposites, green silver nanoparticles, food packaging, food shelf-life, sustainable packaging, starch coating, food waste, biopolymers, green chemistry, colloidal silver

## Abstract

In the pursuit of enhancing food packaging, nanotechnology, particularly green silver nanoparticles (G-AgNPs), have gained prominence for its remarkable antimicrobial properties with high potential for food shelf-life extension. Our study aims to develop corn starch-based coating materials reinforced with G-AgNPs. The mechanical properties were examined using a uniaxial tensile tester, revealing that starch coated with the highest G-AgNPs concentration (12.75 ppm) exhibited UTS of 87.6 MPa compared to 48.48 MPa of control paper, a significant (*p* < 0.02) 65% increase. The assessment of the WVP showcased a statistical reduction in permeability by up to 8% with the incorporation of the hydrophobic layer. Furthermore, antibacterial properties were assessed following ISO 22196:2011, demonstrating a strong and concentration-dependent activity of G-AgNPs against *E. coli*. All samples successfully disintegrated in both simulated environments (soil and seawater), including samples presenting G-AgNPs. In the food trial analysis, the presence of starch and G-AgNPs significantly reduced weight loss after 6 days, with cherry tomatoes decreasing by 8.59% and green grapes by 6.77% only. The results of this study contribute to the advancement of environmentally friendly packaging materials, aligning with the UN sustainable development goals of reducing food waste and promoting sustainability.

## 1. Introduction

Food spoilage imposes a considerable economic burden, contributing significantly to food waste—a leading factor in greenhouse gas emissions, estimated at approximately 10% globally [1]. The UN reported approximately 931 million tonnes of food waste from households, retail, and the food service industry in 2019, accounting for nearly 40% of all food intended for human consumption [2]. The UK government has long prioritised food waste reduction, aiming to slash overall waste by 50% by 2030. This initiative aligns with mitigating economic losses incurred due to food spoilage [1]. Moreover, in line with the objectives set forth by the UK Plastic Pact to significantly diminish or eliminate the usage of key plastic materials, and considering the emphasis on reducing fossil fuel dependency prominently highlighted at COP28, the use of biodegradable materials emerges as a practical and vital solution [3]. Paper and board offer numerous advantages as packaging materials. However, the current technologies used to enhance their barrier properties for food packaging rely on synthetic polymer coatings and lamination with plastic, such as polyethylene (PE), polypropylene (PP), and aluminium foils [4,5,6,7]. These methods not only have a negative impact on packaging sustainability, but also result in poor recyclability and a lack of biodegradability [6,7]. Recently, there has been a growing interest in using biopolymers as coatings for paper, opening up new possibilities for composite formulations that can meet the specific requirements of food packaging [8,9,10,11]. While the research on biopolymers to enhance packaging material barrier properties is on the rise, only a limited number of studies have focused on paper-based materials for food packaging [12]. Polysaccharides such as starch [13], chitosan [14], or pectin [15] stand out as promising candidates to replace synthetic polymers in food paper coatings. They can form a film, exhibit excellent affinity for paper substrates, provide suitable barriers against gases and aromas, and even enhance mechanical strength [11,16,17]. Moreover, these biopolymers are biodegradable, nontoxic, and can serve as a matrix for incorporating additives with specialised functions for coated paper, such as active antimicrobial properties [18,19]. Among them, starch, a versatile biopolymer derived from various plant sources, plays a vital role in food packaging materials [20,21]. Its outstanding film-forming properties, barrier to moisture and gases, compatibility with other biodegradable materials, low cost, and wide availability make it an ideal candidate for creating sustainable packaging solutions [20,21,22]. Starch-based materials offer the potential to improve food preservation, reduce waste, and contribute to eco-friendly packaging alternatives [13,21,22]. However, while polysaccharides are widely employed due to their diverse applications, they often lack intrinsic antimicrobial activity, rendering them susceptible to bacterial growth and necessitating the use of additional active agents [21]. To address these challenges, the integration of nanomaterials has emerged as a promising solution. Specifically designed nanomaterials not only enhance the physicochemical and optical properties of the packaging material, but also confer bioactive functionalities, mitigating the inherent limitations of polysaccharides [11,23,24,25,26,27]. This innovative approach holds great potential for advancing materials with improved antimicrobial characteristics and expanded versatility. Silver nanoparticles (AgNPs), in particular, have gained attention for their strong antimicrobial properties [28,29,30,31]. These nanoparticles exhibit a high surface area and unique physicochemical characteristics that enable them to effectively inhibit the growth of bacteria, fungi, and other harmful microorganisms in food, such as *Escherichia coli* (*E. Coli*), *Enterococcus faecalis,* and *Staphylococcus aureus*, when incorporated into packaging materials ranging from rigid plastics polyethylene (PET) [32], flexible low-density polyethylene (LDPE) [33], paper-based materials, and biopolymers [34]. This innovative approach not only enhances food safety and quality, but also contributes to reducing food waste by prolonging the shelf life of perishable items [11]. Thus, the integration of silver nanoparticles into food packaging represents a promising avenue for improving food preservation and sustainability. However, while biopolymers and AgNPs offer the potential for enhancing food packaging, it is essential to acknowledge the challenges associated with the synthesis and applications of AgNPs. Many traditional synthetic methods involve the use of toxic reducing and stabilising agents such as sodium borohydride (NaBH_4_), raising environmental and health concerns [35,36]. This is where green synthesis methods become crucial. Green synthesis techniques aim to produce silver nanoparticles using eco-friendly and sustainable processes, minimising the use of harmful chemicals, and reducing the environmental impact [36]. In the scope of our study, we employed a straightforward process to fabricate starch-film coatings, integrating AgNPs synthesised through eco-friendly methods onto paper-based packaging. 

In this study, our goal was to develop corn starch-based coating materials reinforced with green-synthesised silver nanoparticles (G-AgNPs) to enhance food shelf-life, reduce pathogenic bacteria growth, and strengthen paper mechanical and water barrier properties of the developed bionanocomposites. The roughness and morphology of the obtained coated paper materials was assessed by means of AFM analysis, which allowed the identification of G-AgNPs within paper surfaces. The characterisation of the mechanical properties of the resulting materials included parameters such as elongation at the break (EaB) and ultimate tensile strength (UTS). The assessment of the water vapour barrier was conducted in accordance with ASTM E96 standards [37], and improvement in paper hydrophobicity was assessed by means of water contact angle (WCA) analysis. Additionally, we investigated the antimicrobial properties of these papers through contact methods, targeting an *Escherichia coli* model strain. Finally, the enhanced food shelf-life of developed materials was investigated in simulated food trials using green grapes, cherry tomatoes, and mushrooms, demonstrating the extended shelf life achieved by incorporating green-synthesised AgNPs into the packaging materials. 

## 2. Results and Discussion

### 2.1. Product Characterisation

#### Silver Nanoparticles Characterisation

The G-AgNPs were characterised using UV–Vis spectroscopy, scanning electron microscopy (SEM), and energy-dispersive X-ray spectroscopy (EDS).

UV–Vis spectroscopy was used to confirm the formation of AgNPs, and the spectrum is shown in Figure 1. The characteristic peak of Metalchemy’s G-AgNPs is at ~413 nm, which corresponds to the surface plasmon resonance (SPR) of AgNPs, observed in other synthesis processes [17]. The intensity of the peak was directly proportional to the concentration of G-AgNPs in the solution, where a 20 ppm suspension resulted in an absorbance of 0.85.

SEM images showed the morphology and size distribution of the G-AgNPs. Figure 2 shows randomly selected areas at different magnifications. These revealed that the G-AgNPs were uniformly distributed and spherical in shape with a diameter range of 5 to ~20 nm, with a mean value of 19 nm. The high-resolution SEM images showed that the surface of the G-AgNPs was smooth and free from any impurities.

EDS analysis was performed to confirm the elemental composition of the G-AgNPs, and is shown in Figure 3. The EDS spectrum showed a strong peak for silver, indicating the presence of AgNPs. The other peaks in the spectrum corresponded to aluminium originating from the holder stub.

The results of the characterisation analysis showed that the synthesised G-AgNPs were spherical in shape, with an average diameter of the silver core of 19 nm. The UV–Vis spectrum confirmed the presence of G-AgNPs, while the SEM images revealed their morphology and size distribution. EDS analysis confirmed the elemental composition of the G-AgNPs. The results of DLS analysis evidenced the presence of sharply distributed AgNPs with a mean diameter of 95.8 ± 2.3 nm with a polydispersity index (PI) of 0.3, confirming the presence of macromolecules coating and surrounding the silver core. The particle size distribution in function of intensity % is reported in Figure 4. Indeed, DLS analysis allows the determination of hydrodynamic volume of NPs composed by the inorganic core and the stabilising agent. G-AgNPs used in this study were produced with a green plant extract-mediated method of synthesis based where macromolecules such as polysaccharides and proteins stabilised the formed AgNPs. The presence of a coating surrounding metal nanoparticles is fundamental. It ensures high stability due to the steric hindrance provided by the macromolecules [36]. Additionally, it maximises possible interactions, such as hydrogen bonds, within the AgNPs and the polymeric materials once internalised within thermoplastic matrices [31]. 

### 2.2. Paper-Based Packaging Characterisation

#### 2.2.1. Starch-Based Coating FTIR Analysis

Starch-modified samples evidenced the OH, C-H, C-O-C, and C-O functional groups, respectively. Furthermore, the characteristic C-O-C ring vibration in starch leads to an absorbance peak at around 700–900 cm^−1^ not detected in the paper control sample (Figure 5). The C-O bending associated with the OH group causes an absorbance peak at around 1648 cm^−1^.

Moreover, the absorbance peak at 1415 cm^−1^ observed in the starch-modified samples confirmed the presence of C-H symmetrical scissoring of the CH_2_OH moiety [38].

#### 2.2.2. Morphological Properties of Paper-Based Packaging

The roughness of coated papers has been measured by means of AFM analysis. The presence of AgNPs did not affect the roughness of starch-coated materials with an average value of 80.2 ± 12.2 nm for StarchPaper_G-AgNPs_10%, and 77.9 ± 8.6 nm for control with starch (Figure 6). Starch mainly contributes to the material roughness, while the AgNPs, due to the comparable size of the roughness, did not contribute to a statistically increased value. However, AFM allowed the detection of G-AgNPs on the paper surface not detected in the control (Figure 6). The revealed diameter of detected AgNPs was 45.2 ± 6.2 nm. The higher diameter detected by AFM compared to SEM images can be due to the G-AgNPs interacting with starch, which can act as a further stabilising agent of the AgNPs, resulting in a larger size.

#### 2.2.3. Mechanical Properties of Paper-Based Packaging

The assessment of mechanical properties of the developed paper-based coated materials is integral to determining the material’s fitness for physically safeguarding food throughout manufacturing, transit, and storage. Two pivotal parameters, elongation at break (EaB) and ultimate tensile strength (UTS), were measured to gauge the material’s performance. A higher elasticity, combined with robust tensile strength, is indicative of superior material quality [39,40]. Table 1 provides an overview of the mechanical properties of coated papers. Surprisingly, the introduction of G-AgNPs into the paper coating revealed a significant enhancement in the EaB% of the resulting material (Table 1). This finding appears contrary to the existing literature, where the presence of metal nanoparticles, particularly silver, is commonly associated with a typical reduction in elongation at break by 2–3% [41]. This result could be ascribable to the complex interaction between the stabilising agents and the polymeric matrix of starch coating, such as charge-charge and hydrogen bonds, which can affect the mechanical properties of the material. Compared to the reported literature, which commonly use synthetic-polymer based capping agent, G-AgNPs are stabilised by a large natural polymeric coating, which can maximise the interaction with starch chains resulting in a significant increase (*p* < 0.05) of EaB values compared to the control. Indeed, the EaB% increased from 5.05% ± 1.85% in the control paper to 8.34% ± 1.66% in the sample with the highest silver concentration (StarchPaper_G-AgNPs_30%).

The UTS results exhibited a fluctuating up-and-down pattern with increasing concentrations of AgNPs (Table 2). Interestingly, the most noteworthy increase was observed in the StarchPaperControl (absence of AgNPs) with a value of 110.54 ± 18.01 MPa, confirming the contribution of starch integration in the paper improving its mechanical properties [42,43]. This phenomenon exhibited inconsistency upon the introduction of AgNPs, suggesting a complex interplay that could hinder the strengthening process.

**Table 1 polymers-16-00941-t001:** Mechanical properties of samples.

Sample	EaB%	UTS (Mpa)	EaB% Data Literature [44,45]
PaperControl	5.05 ± 1.85	48.48 ± 5.3	
StarchPaper	3.19 ± 0.61	110.54 ± 18.01	7.6 ± 2.5 *
StarchPaper_G-AgNPs_5%	6.97 ± 2.30	68.38 ± 35.68	6.8 ± 2.0/10.2 ± 2.5 **
StarchPaper_G-AgNPs_10%	7.14 ± 1.35	45.25 ± 8.17	
StarchPaper_G-AgNPs_20%	7.62 ± 2.46	67.01 ± 26.33	
StarchPaper_G-AgNPs_30%	8.34 ± 1.66	87.63 ± 24.23	

* Data reported in the literature for starch surface functionalised materials; ** data reported in the literature for starch-AgNPs surface functionalised materials; results obtained with a minimum of 10 repetitions per sample (*p* value < 0.05).

Indeed, varying concentrations of AgNPs did not yield a linear change in UTS results, as reported in Table 2. Nevertheless, the sample with 30%AgNPs concentration presented a 1.81-times increase in tensile strength compared to the control base paper packaging. Further investigation is required to elucidate the intricate dynamics between polysaccharides and AgNPs and their collective impact on UTS in paper-based materials. The positive mechanical property enhancements observed in our study suggest that the incorporation of starch/G-AgNPs into paper coatings has the potential to elevate packaging mechanical performance. These improvements could lead to more resilient and durable packaging materials, addressing critical challenges in food preservation. In food packaging, flexibility is a fundamental parameter to avoid bioplastic tears or breaks during handling and transportation, critical factors in ensuring the safety, freshness, and quality of the packaged food products.

#### 2.2.4. Water Vapour Permeability (WVP) and Water Vapour Transmission Rate (WVTR)

WVP is a main packaging property that indicates resistance capability to water vapour transmission of packaging materials. A low WVP is required to minimise moisture transfer from the surrounding environment to the packaged product, particularly under humid conditions. The addition of starch as a coating onto the paper decreased (*p* < 0.05) the WVP by 2.3% (Table 3). Examining the effects of varying concentrations of AgNPs, samples with lower AgNP concentrations, such as StarchPaper_G-AgNPs_5%, StarchPaper_G-AgNPs_10%, and StarchPaper_G-AgNPs_20%, displayed slight deviations in WVP from the sole starch-coated samples.

For instance, StarchPaper_G-AgNPs_5% exhibited a WVP of approximately 122.64 g/s·m·Pa, marginally lower than the sole starch-coated sample (127.92 g/s·m·Pa) (Table 2).

In contrast, the highest tested concentration of AgNPs (StarchPaperG-AgNPs30%) showcased a considerable decrease (*p* < 0.05) in WVP, with the WVP dropping notably by 8.6%, indicating a more pronounced impact on moisture permeation compared to the other AgNP concentrations due to the higher hydrophobicity of the material. This enhancement in the moisture barrier should provide increased protection against degradation, spoilage, and potential contamination, ensuring the integrity and safety of the packaged product [21].

The obtained results are aligned with the literature of starch-based coatings, where the WVP is commonly 1–3 × 10^−6^ g/s·m·Pa [46,47]. Nevertheless, a further decrease is required if it is to compete with synthetic flexible packaging materials such as LDPE, with WVP ranging between 10–11 and 10–12 g/s·m·Pa [46,48].

#### 2.2.5. Hydrophobicity

An effective food packaging material must inherently exhibit hydrophobicity to shield its contents from water. Previous studies underscored the significance of increased hydrophobicity in mitigating the rate of oxidation and deterioration, thereby contributing to a reduction in the loss of food quality and flavour [49]. In this study, starch-coated paper-based packaging demonstrated significantly higher WCA values than the control; the latter exhibited a WCA of 82.3° ± 5.3°, while the starch-coated film increased the angle by 10.4% to 86.9° ± 2.5°. With a significant AgNP concentration, the material’s hydrophobicity statistically increased by 18%, 25%, 25%, and 32% (*p* < 0.05) (Table 3). This can be attributed to the presence of AgNPs within the layer, and is in line with observations by Roy and Rhim, who documented improved hydrophobicity in starch/agar/AgNP bionanocomposites [50].

#### 2.2.6. Thermal Analysis

TGA thermograms of the starch paper nanocomposites are shown in Figure 7, and the thermal degradation characteristics are given in Table 4. The thermograms show that the addition of nanoparticles affects the thermal stability of the nanocomposites and decrease it to the lower temperature. Although Rozilah et al. reported that the addition of silver nanoparticles to sugar palm starch bionanocomposites resulted in improved thermal properties due to the heat stability of silver nanoparticles [51], in our study, the observed results were different, which may be due to the multilayer material and the integration of silver nanoparticles into the bionanocomposite structure. It should be noted that the presence of a higher concentration of silver nanoparticles resulted in a higher weight residue due to the thermal stability of the silver nanoparticles.

The melting temperature of the starch paper nanocomposites was determined from the DSC heating thermograms, which are shown in Figure 8 and Table 5. The results of DSC thermograms did not indicate the glass transition temperature (Tg) for the bionanocomposite films.

The melting temperatures of nanocomposites are reported in Table 5, and as can be seen, the melting point of the films was affected by the presence of silver nanoparticles. The values decreased when the film was developed in the higher concentration of silver nanoparticles. This behavior might be related to the high thermal conductivity of silver nanoparticles that leads to enhancement of the thermal conductivity of the films and reduces the melting temperature. Similar results were reported by Kumar [52].

#### 2.2.7. G-AgNP Migration

Migration tests are used to check the inertia of packaging suitable as food contact material by analysing the possible leaching of components from the packaging material to food. These migration tests are carried out using acetic acid as food simulants. Migration testing quantifies the transfer of molecules from food contact materials into food. Quantifying this parameter is mandatory for regulatory approval of active packaging [44,53]. The European Food Safety Authority (EFSA) has defined a maximum limit of Ag^+^ migration from active packaging permissible in food to be 0.05 mg/kg in food and 0.05 mg/L in water [11]. The migration of Ag^+^ ions of the sample with the highest silver concentration was assessed. The ICP-MS analysis of the G-AgNP migration from StarchPaper_G-AgNPs_30% paper-based packaging showed that the tested leachates had almost undetectable levels of Ag^+^ (<0.006 mg/kg). Overall, the results indicate that silver migration was well below the maximum threshold set by the EFSA. Hence, the developed novel material meets the safety migration levels set by EFSA for food packaging materials.

### 2.3. Paper-Based Packaging Antimicrobial Activities

The results of antimicrobial activity in terms of the zone of inhibition are reported in Table 6. The presence of G-AgNPs increased the inhibition area, with the most pronounced and statistically significant (*p* < 0.002) effect observed at a 30% concentration of G-AgNPs (0.207 cm) after 48 h. Notably, the lowest G-AgNPs concentration (StarchPaper_G-AgNPs_5%) exhibited antimicrobial activity after only 48 h.

The obtained results align with the existing literature, where the incorporation of AgNPs in composite films in concentrations ranging from 5 to 50 ppm consistently demonstrates an inhibition zone of approximately 0.15–0.2 cm against *E. coli* [54,55].

### 2.4. Bio-Disintegration of Starch Coated Papers

Bio-disintegration and degradation of the starch paper materials with different concentrations of AgNPs were studied in soil and simulated seawater over a period of 60 days using low density polyethylene (LDPE) films as nonbiodegradable and non-bio-disintegrable reference materials.

#### 2.4.1. Soil Bio-Disintegration

The results of soil bio-disintegration weight loss and pictures are reported in Table 7. The results obtained of the starch paper materials weight losses during the degradation period show that more than 15% of the paper was degraded after 7 days, regardless of the G-AgNP concentrations. From the visual images of the papers, the degradation and disintegration process of the papers started after 7 days, and after 30 days they lost their initial visual integrity and were pulverized and impossible to weigh (Table 7). This result confirmed that the presence of starch, as well as G-AgNPs even at highest concentrations, did not affect the bio-disintegration process in soil of the coated materials.

#### 2.4.2. Simulated Seawater Bio-Disintegration

Paper materials, besides their inherent biodegradability and safety, can persist in water for long periods [18,19]. Contrary to soil degradation, in seawater, paper coated materials demonstrated slower degradation compared to the pristine paper without starch; however, no differences were observed in the presence or absence of G-AgNPs (Figure 9). After 60 days, the uncoated paper showed a reduction of the starting weight of 30%, while for starch coated papers (in presence and/or absence of G-AgNPs), a statistical reduction (*p* < 0.05) of 15% circa was observed. This behavior could be the result of the higher hydrophobicity showed by starch-coated materials demonstrated in the present study, making the paper cellulose less accessible to seawater microorganisms.

### 2.5. Food Trials

Among the obtained samples, StarchPaper_G-AgNPs_10% was selected for further characterisation. This selection was made based on its robust antimicrobial activity and the concurrent enhancement of barrier properties achieved with a low concentration of silver. While higher concentrations exhibited slightly greater antimicrobial activity, the marginal increase was not deemed sufficient to justify their use, especially considering the doubled or tripled amounts of silver involved, which would cause an increase in cost. This choice positioned it as a promising candidate for food trial experiments. Three different foods, namely, cherry tomatoes, green grapes, and mushrooms, were used as a case study to assess the reduction of weight loss and food discoloration by means of a hue test.

#### 2.5.1. Weight Loss

Weight loss is a parameter used for measuring the rate of food spoilage [56]. In an open system, the degradation of food is influenced not only by internal chemical processes, but also by interactions with the external environment. Barrier permeability of food packaging materials plays a crucial role in regulating the exchange of gases, particularly oxygen and water vapour, between the food product and the surrounding atmosphere [57]. When packaging materials exhibit higher permeability to water vapour, moisture from the food can readily escape into the environment. Likewise, if the packaging allows oxygen to penetrate, oxidative reactions can occur within the food matrix, leading to the breakdown of complex molecules [56]. Both of these processes contribute to the release of smaller molecules, such as amino acids, glucose, and fatty acids, into the surrounding environment, resulting in the observed weight loss of the food product. The weight loss data allow for effective safety and quality assessments of food shelf-life for a variety of storage conditions and packaging materials [52]. The results presented in Figure 10 of the most promising sample (StarchPaper_G-AgNPs_10%) highlight the efficacy of the developed paper-based packaging in reducing the rate of weight loss for cherry tomatoes and green grapes, commencing from day 1. A statistically significant reduction in weight was observed, amounting to 8.59% (*p* < 0.05) for cherry tomatoes and 6.77% (*p* < 0.05) for green grapes at day 6, compared to the control. Conversely, no improvement of mushroom weight loss was observed after 6 days.

#### 2.5.2. Hue Test

The hue test, as a colour attribute, provides valuable information about the perceptual quality and freshness of food items. It is a measure of the dominant wavelength in the colour spectrum and is closely associated with the overall colour appearance. Therefore, changes in hue can be indicative of alterations in the visual appeal and freshness of the food.

StarchPaper_G-AgNPs_10%-coated cherry tomatoes and green grapes demonstrated statistically (*p* < 0.05) higher hue values compared to the control, suggesting a potential better preservation of colour intensity throughout the storage period (Figure 11). The results of colour analysis have been reported as hue angle, and can be observed in Figure 12. The results revealed that all samples displayed a progression towards browning from day 0 to day 6, resulting in a decrease of hue value due to the colour variation. This phenomenon can be used in open packaging; it is plausible that no significant differences were recorded or attributed to the enzymatic action of polyphenol oxidase (PPO), which reacts with oxygen to produce melanins, leading to brown discoloration in fruits and vegetables [45].

## 3. Discussion

The incorporation of G-AgNPs led to improvements in all paper characteristics that could not be achieved through the use of starch alone. Additionally, the AFM analysis demonstrated that the inclusion of G-AgNPs did not impact the roughness of starch-coated materials; the average roughness values were 80.2 ± 12.2 nm for StarchPaperG-AgNPs10% and 77.9 ± 8.6 nm for the control with starch. The mechanical properties of the materials showed improvement, with up to a ×1.65 increase in EaB% observed with higher AgNP concentration. This trend was also evident in the water contact angle (WCA) and water vapour permeability (WVP) analyses, where the increase in AgNP concentration led to a higher hydrophobic behaviour of the materials. While the starch coating displayed a modest yet noteworthy decrease in WVP, indicating a promising approach to enhance the paper’s moisture resistance, the presence of G-AgNPs on WVP resulted in a significant increase of WVP and WVTR values to 117.9 (g/m^2^/24 h) and 2.03 × 10^−6^ (g/s·m·Pa), respectively, for the sample with highest concentration of G-AgNPs. This can be the result of AgNP interactions with the starch polymeric matrix and with paper cellulose. Indeed, AgNPs present a high affinity with macromolecules such as polysaccharides due to the large number of charged chemical groups within polymer repeating units; these interactions, such as hydrogen bonds and charges, can strongly affect the AgNP interactions with water molecules [36]. Moreover, the highest WCA value (110.0° ± 3.5°) was obtained using 30% AgNPs, which is comparable to highly hydrophobic systems such as polydimethylsiloxane (PDMS) and polyethylene, which commonly exhibit WCAs above 110°. The observed increase in WVP and WCA, indicating heightened surface hydrophobicity, can be primarily ascribed to the inherently hydrophilic nature of the nanoparticles. Additionally, this phenomenon can be attributed to the formation of intermolecular hydrogen bonds between the active surface functional groups of the nanoparticles and the polymer matrices, contributing to a reduction in hydrophilic groups on the film surface [50].

The migration test conducted on the sample with the highest AgNP concentration demonstrated the safety applications of the developed formulations, with migration levels below 0.0006 mg/kg of food. This is 1.67-times lower than the 50 µg/kg level prescribed by EU and UK regulations. Silver, albeit in trace amounts, is naturally present in everyday foods, with adults consuming an estimated 20–80 µg per day. Concern arises over the potential migration of silver from packaging into food, which could elevate dietary exposure. Accurate quantification of silver migration becomes imperative in this context. To address this, regulatory bodies such as the European Food Safety Authority (EFSA) and the United States Food and Drug Administration (USFDA) have established guidelines governing the use of silver nanoparticles (AgNPs) in active packaging materials. The EFSA stipulates that AgNPs should not surpass 0.05 mg/L in water and 0.05 mg/kg in food [12]. Analysing the migration profile of silver is crucial for ensuring the maintenance of its antibacterial efficacy while adhering to prevailing regulations.

Notwithstanding the low migration, G-AgNP starch-coated papers exhibited strong antimicrobial properties compared to using only starch, indicating a concentration-dependent activity of G-AgNPs. The mechanism of action of AgNPs against pathogenic bacteria has not yet been well elucidated, since it is a process which occurs by means of several complex pathways.

Bio-disintegration studies confirmed that the presence of G-AgNPs did not affect the degradation profile in both simulated environments, demonstrating identical bio-disintegration of starch paper materials. In simulated seawater, starch-coated materials demonstrated a slower bio-disintegration compared to pristine paper. This behaviour could be the result of higher hydrophobicity showed by starch-coated materials demonstrated in the present study, making the paper cellulose less accessible to seawater microorganisms.

The preliminary food trials on packaged mushrooms, cherry tomatoes, and green grapes confirmed improvements in reduced weight loss, appearance, and retention. The presence of G-AgNPs resulted in a statistically significant decrease in food weight loss after 6 days, with reductions of 8.59% for cherry tomatoes and 6.77% for green grapes. Conversely, no improvement of mushrooms weight loss was observed after 6 days. This could be attributed to the fact that the mushrooms were halved before being packaged. This could have impacted the respiration, breakdown, and oxidation rates of mushrooms within the open packaging, potentially diverging from the observed behaviour in the other fruits under examination [45,57,58]. Considering the above, further investigations are warranted to elucidate the effects of the cutting process on the shelf-life of mushrooms. The improvement in food appearance was further confirmed by the hue test, with slower colour variation to brown for both samples.

## 4. Materials and Methods

### 4.1. Materials

Corn starch was purchased from Sigma-Aldrich (Manchester, UK). The paper-based packaging was obtained from APEX CE Specialists Ltd. (Manchester, UK). The G-AgNPs (green synthetised AgNPs aqueous suspension) were produced in-house at Metalchemy’s laboratory (London, UK). HPLC-grade water, tryptone, yeast extract, sodium chloride, and agar were purchased from Sigma-Aldrich (UK). Soil was purchased from Westland, UK.

### 4.2. AgNPs Characterisation

#### 4.2.1. AgNPs UV–Vis Analysis

To analyse the G-AgNP solution and measure the amount and dispersion of nanoparticles, a Shimadzu UV-1900i spectrophotometer (Tokyo, Japan) was used to record UV–Vis spectra, scanning the absorption spectra in the 300–700 nm wavelength range.

#### 4.2.2. Scanning Electron Microscopy (SEM) with Energy Dispersive Spectrometry (EDS)

The AgNP morphology was assessed by means of an SEM instrument (Zeiss EVO HD with eBruker EBSD, Jena, Germany) operated with an accelerating voltage of 10 kV and working distance (WD) of 8.3 ± 1 mm. The 14 mm circular modified samples were fixed on conductive carbon tapes and coated with 95% gold and 5% palladium using a Polaron E5000 Sputter Coater (Quorum Technologies, Laughton, UK). Images of the samples at magnifications of 100k, 30k, and 5k× were taken to visualise the samples. The AgNP average size was calculated using ImageJ software (https://imagej.net/ij/download.html, 23 March 2024). At least 50 nanoparticles were selected from different acquired images.

#### 4.2.3. Particles Size Distribution

The size of AgNPs and their polydispersity index (PDI) were determined using dynamic light scattering (DLS) with a Zetasizer Nano-ZS (Malvern Instruments, Manchester, UK; model: ZEN3600, serial number: MAL1000973) and disposable cuvettes (Fisherbrand, Loughborough, UK; FB55147). Prior to measurement, each sample was filtered by means of a 0.22 µm PTFE filter. After filtration, these samples were measured without dilution and with 10-fold dilution using ultrapure HPLC-grade water (Sigma Aldrich, Gillingham, UK; CAS No: 7732-18-5; PCode: 102604938; Source: BCCK3219). Each sample was measured three times at 25 °C, and the mean ± standard deviation values were calculated.

### 4.3. Preparation of Starch/Green Silver Nanoparticles (G-AgNPs) Paper-Based Packaging

#### 4.3.1. Starch/G-AgNPs, Nanocomposite

To prepare the Starch/G-AgNPs composite coating, 8 g of starch was dissolved in 100 mL of deionized water at 90 °C while subject to vigorous mechanical stirring for 30 min. To prepare starch-AgNPs nanocomposites, different concentrations of G-AgNPs were then added to the starch solution, as detailed in Table 8.

#### 4.3.2. Starch/G-AgNPs Deposition on Paper-Based Packaging

The Starch/G-AgNPs solution at 65 ± 3 °C was deposited on the paper-based packaging using a syringe with a ratio of 1 mL: 76 cm^2^. The bar coating method was applied for the coating, where a metal bar is used to spread the deposited bionanocomposite uniformly over the packaging. The coated samples were left to air dry at room temperature (25 °C).

### 4.4. Paper-Based Packaging Characterisation

#### 4.4.1. Morphological Characterisation—Atomic Force Microscopy

The surface topography of the StarchPaperG-AgNPs10% and Starch Paper Control with only starch coating was evaluated using an atomic force microscope (NaioAFM NanoSurf, UK). For each sample, the scans were carried out on an area of 48 × 48 μm at 128 (lines) × 128 (dots) resolution. The roughness parameter (Rq, root mean square) was calculated with the data of the topographic micrographs, using the equipment’s software. To obtain more accurate values to determine the presence of AgNPs within coated papers, scans were carried out on an area of 1.5 × 1.5 μm at 256 (lines) × 256 (dots) resolution. AgNP size was determined using the equipment’s software. At least 10 nanoparticles were selected from different acquired images.

#### 4.4.2. Mechanical Testing

A universal testing machine from AML Instruments (Lincolnshire, United Kingdom) was used to determine both the ultimate tensile strength (UTS) (N/mm^2^) and elongation at break (EaB) (%) of the paper-based packaging and Starch/G-AgNPs paper-based packaging samples. The samples were cut into dog-bone shapes that were 5 mm wide and 38 mm long with an American Society for Testing and Materials (ASTM) D412 knife mould rubber knife [59]. Each test sample was placed into the grips of the testing apparatus, which were set 30 mm apart, and tested to failure at a speed of 30 mm/min. Prior to testing, the thickness of each sample was measured using a digital vernier calliper (±0.001 mm). Stress/strain curves were generated for each material using the collected data. Results were obtained with a minimum of 10 repetitions per sample. The stress (N/mm^2^) for each material was calculated using the data from these curves, which was then exported into a Microsoft Excel, Version 2109 (Build 14430.20270) spreadsheet.

The *UTS* was calculated using Equation (1):(1)$$UTS=\frac{\;F_{MAX}}{A_{0\;}}$$
where *UTS* = ultimate tensile strength, *F_MAX_* = maximum force (N), and *A*_0_ = cross-section area ($mm^{2}$).

*EaB* was calculated using Equation (2):(2)$$EaB=\frac{\;\Delta L\;}{L}*100$$
where *EaB* = elongation at break, *L* = initial length (mm), Δ*L* = change in final length and initial length (mm).

#### 4.4.3. Water Vapour Permeability (WVP)

In accordance with ASTM E96, a gravimetric modified cup method was used to assess the water vapour permeability (*WVP*) [60]. Plastic test tubes (50 mL) were filled with distilled water to a level 3⁄4 (35–40 mL) with a distance of circa 2 cm from the paper samples that were applied to the vials by means of film layer. For each vial, the initial weight was calculated using a weighing balance. The vials were then placed in the controlled chamber, which also contained a beaker of water. Then, vials were weighed every day for two weeks. A linear regression analysis of time vs. volume change (R^2^ > 0.99) was used to determine the slope of each line. Equations (3) and (4) were used to calculate the water vapour transmission rate (*WVTR*) and *WVP*, respectively.
(3)$$WVTR=\frac{Slope}{A_{sample}}$$
(4)$$WVP=\frac{G}{t*A}$$
where *G* is the weight change in grams; *t* is the duration of the test in hours; *A* is the test area in m^2^.

During all WVP measurements, air surrounding the membranes had a constant temperature of 24 °C and relative humidity of 70%. Results were obtained with a minimum of 3 repetitions per sample.

#### 4.4.4. Antimicrobial Activity

The disk diffusion method (ISO 22196:2011 [61]) was used to determine the antimicrobial activity of starch paper nanocomposites against *E. coli* strain (ATCC 2592). A volume of 0.4 mL of 108 cells/mL *E. coli* grown for 24 h in 20 mL of Luria broth (LB) (tryptone 10 g, yeast extract 5 g, NaCl 10 g, 1 L deionised water) nutrient agar was dropped onto LB nutrient agar media (tryptone 10 g, yeast extract 5 g, NaCl 10 g, 15 g agar, 1 L deionised water), and then spread on the entire surface of the plate using a sterile spatula. Subsequently, sterile squares were carefully applied onto the surface of the nutrient agar plate using sterile swabs. Afterwards the plates were incubated in a Sciquip Incubator S-Series SQ-4615 at 37 °C. The inhibition zones around the discs where no growth occurred were measured in millimetres, by recording the pictures after 24 and 48 h with a camera and conducting data analysis with ImageJ software. Experiments were conducted in triplicate for each sample. At regular intervals, pictures of each Petri dish were taken, and the inhibition zone was tracked using ImageJ software.

#### 4.4.5. Hydrophobicity

To compare the hydrophobicity of the G-AgNPs/Starch paper-based packaging surfaces, the water contact angle (WCA) of each sample was determined. A 10 µL drop of DI water was added onto the sample surfaces. Images were recorded at a 90° perspective. Contact angles were measured using ImageJ software and the results were presented as an average of left and right contact angles. Results were obtained with a minimum of 10 repetitions per sample.

#### 4.4.6. Migration of G-AgNPs

The G-AgNP migration from coated paper was assessed in accordance with the requirements of Article 11 of Regulation (EC) No 882/2004. The sample coated with the highest AgNPs concentration (StarchPaperG-AgNPs30% sample) was selected for the analysis as a case study. Briefly, the StarchPaperG-AgNPs30% sample paper was submerged in 3% acetic acid solution in a ratio of 0.02 dm^2^/mL. Analysis was performed in triplicates. Samples were stored at 60 °C for 10 days. Inductively coupled plasma mass spectrometry (ICP-MS) was used to analyse the concentration of migrated silver (μg/kg).

#### 4.4.7. Thermal Analysis

Thermogravimetric analyses of starch paper nanocomposites were carried out with a TGA 178 Q50 (TA Instrument, New Castle, DE, USA). Around 2 mg of each sample was placed in an aluminium pan and heated from 20 °C up to 550 °C at a rate of 10 °C/min, under nitrogen at 40 mL/min flow rate.

Differential scanning calorimetry (DSC) analyses were performed using a DSC Q2000 Differential Scanning Calorimeter (TA Instruments) in a nitrogen gas atmosphere. Samples weighing approximately 4 mg were used for DSC measurements, and were placed in a standard aluminium pan and the maximum ramp temperature was set at 170 °C starting from 10 °C, at a rate of 10 °C /min with isothermal for 5 min. After an isothermal phase, the samples were cooled down to −10 °C at a rate of 10 °C/min. After that, a second heating ramp up to 170 °C from 10 °C at a rate of 10 °C/min was recorded.

#### 4.4.8. Fourier Transform Infrared Spectrometer (FTIR)

The infrared spectrum of all samples was determined using an attenuated total reflectance Fourier transform infrared spectrometer (ATR-FTIR) (Spectra Science Ltd., Chesham, Bucks, UK) equipped with a ZnSe attenuated total reflectance (ATR) crystal accessory at room temperature. The spectra of the starch paper nanocomposites were acquired in the range of 4000–650 cm^−1^ with a resolution of 4 cm^−1^ and 32 scans per sample.

### 4.5. Bio-Disintegration

Bio-disintegration of the developed materials in the soil was evaluated by burying the paper pieces (1 cm × 1 cm) under soil. The soil was poured into plastic trays (5 cm × 5 cm × 5 cm) up to a height of circa 3 cm. The buried papers were kept at room temperature (25 °C) for 60 days, and 1 mL of deionised water was sprayed on the top of the soil every 7 days. From the beginning of the experiment, the weight of the papers was recorded at different times (0, 7, 30, and 60 days). The percentage of paper bio-disintegration was calculated as the percentage of paper weight loss compared to initial paper weight. The experiment in simulated seawater was performed in a similar way, soaking paper pieces (1 cm × 1 cm) in 40 mL of seawater in 50 mL plastic tubes. Before weight, samples were washed by soaking the pieces in 100 mL of deionized water for 10 s to remove the salts.

### 4.6. Food Trials

#### 4.6.1. Packaging Food Samples

Whole green grapes, cherry tomatoes, and halved mushrooms from a local market in London, UK, were packaged in Starch/G-AgNP paper-based packaging and control paper-based packaging for a total of 18 samples. The bags packaged one sample (green grape/cherry tomato/half mushroom) individually and were created with three sides sealed and one side open. The samples were stored at 25 °C and 66% humidity for a duration of 6 days.

#### 4.6.2. Weight Loss

The food sample’s weight was recorded each day of the trial’s duration, using a microbalance (Kern ADB 200 1 ± 0.0001 g). Each sample was measured in triplicate, and then mean values were calculated for analysis for a total of 18 samples.

#### 4.6.3. Colour Measurement—Hue Test

The purity or intensity of the colour (*Hue*) per sample was recorded with a camera and the picture analysed with MatLab software (https://uk.mathworks.com/support/dws.html, 14 March 24); the colour intensity was calculated using Equation (5).


(5)
$$\mathrm{Hue}=(a^{2\;}+b^{2}{)}^{0.5}$$


### 4.7. Statistical Analysis

Statistical differences were analysed using a one-way analysis of variance (ANOVA), and a Tukey test was used for post hoc analysis. A *p*-value < 0.005 was considered statistically significant.

## 5. Conclusions

In conclusion, the integration of G-AgNPs represents a novel and cost-effective strategy for the development of advanced materials for food packaging, opening the door to the green manufacturing of materials with unique properties, further optimised through the use of novel nanomaterials. Specifically, the use of G-AgNPs synthesised through a green and sustainable method has enabled the advancement of paper-based materials for food packaging without the need for harsh methodologies, relying instead on environmentally friendly processes. This integration has led to an enhancement of various paper features, contributing antimicrobial activities that are not achievable through the use of only starch. Future studies should focus on the investigation of various classes of foods nutritional analysis, as well as assessing the environmental (LCA) and techno-economic impact of the novel bionanocomposite packaging material compared to conventional food packaging, such as polyethylene coated paper-based packaging. The aim of future studies will be to evaluate the reduction of food waste and improvements in the shelf-life of different food categories, including meat and fish, aligning future studies with UN sustainable development goals. The results of this study suggest promising applications across a wide range of paper-based materials commonly used in food packaging, particularly in precooked foods, where cardboard-based packaging is prevalent. Moreover, we believe that our research can contribute to achieving United Nations Sustainable Development Goal (SDGs) 2 (end hunger, achieve food security and improved nutrition, and promote sustainable agriculture). Indeed, reducing food waste is a fundamental concern, with much of it attributed to food spoiling, which our technology aims to address. Many foods, particularly fresh fruits and vegetables, require the use of preservatives, some of which can be harmful to health. The antimicrobial activity of our packaging material could help limit the need for such preservatives. Further research is needed to fully explore the potential of starch coatings on various biodegradable materials commonly employed in food packaging, thereby expanding the application possibilities of this innovation.

## Figures and Tables

**Figure 1 polymers-16-00941-f001:**
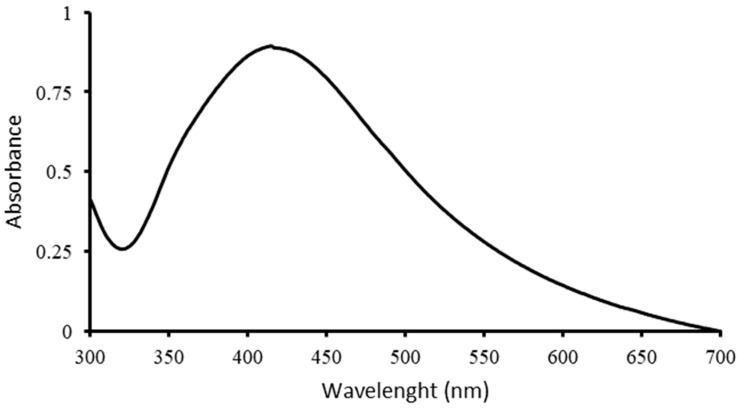
UV–Vis spectroscopy of G-AgNPs at a concentration of 20 ppm.

**Figure 2 polymers-16-00941-f002:**
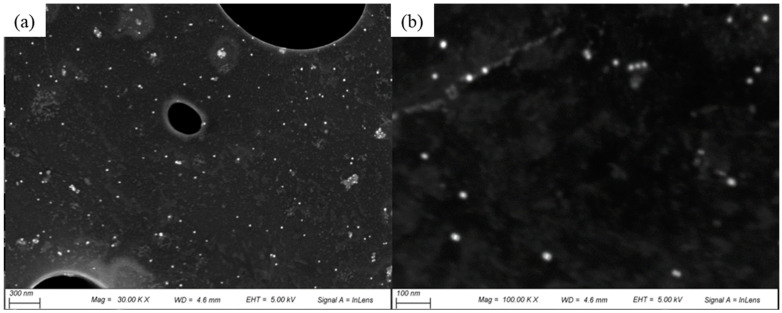
SEM images of G-AgNPs obtained at (**a**) 10.0 k and (**b**) 100.0 k magnification.

**Figure 3 polymers-16-00941-f003:**
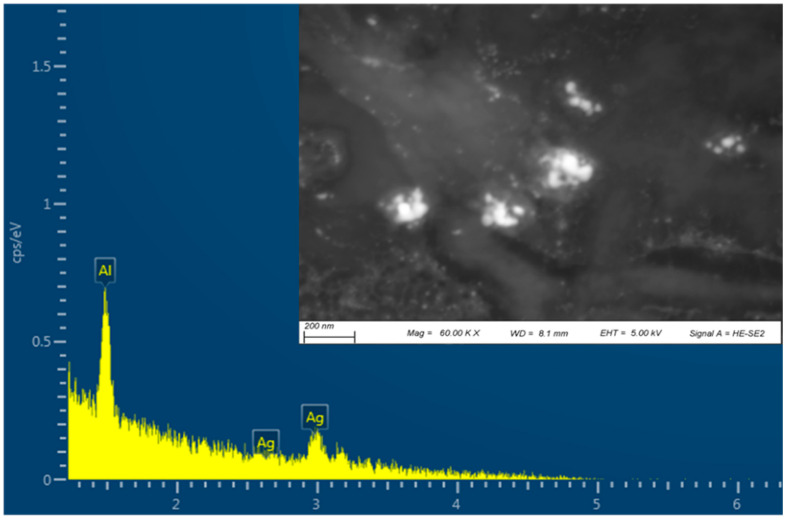
Scanning electron microscopy (SEM) and EDS mapping of G-AgNPs.

**Figure 4 polymers-16-00941-f004:**
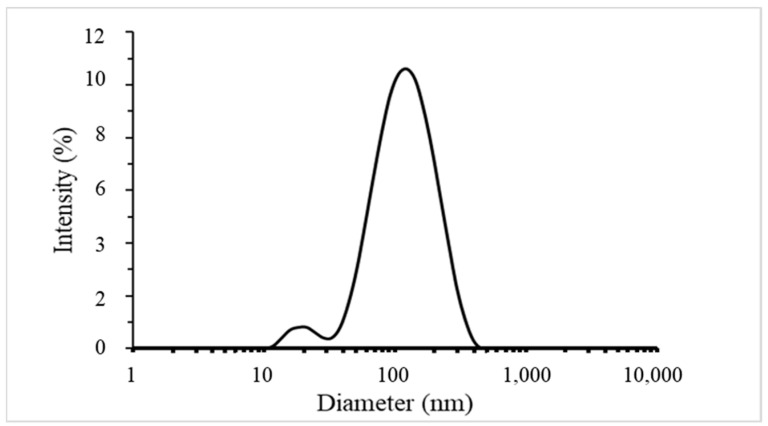
DLS size distribution of G-AgNPs by intensity.

**Figure 5 polymers-16-00941-f005:**
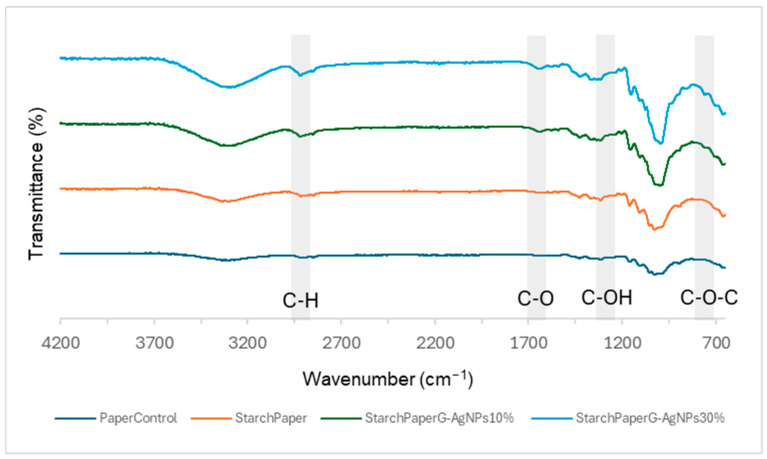
FTIR spectra of starch paper nanocomposites and control paper.

**Figure 6 polymers-16-00941-f006:**
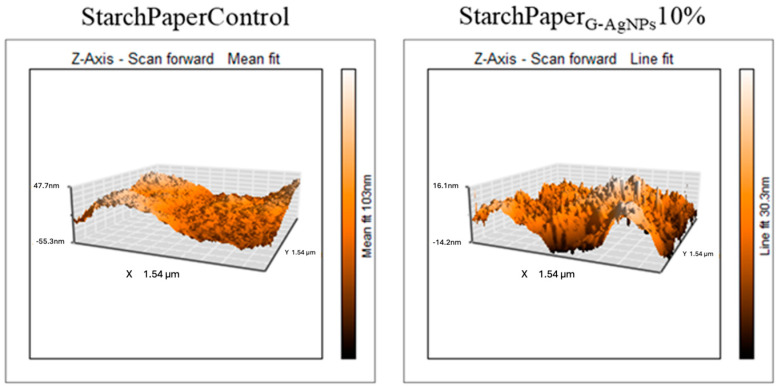
AFM topographic images of starch coating in the presence and absence of AgNPs.

**Figure 7 polymers-16-00941-f007:**
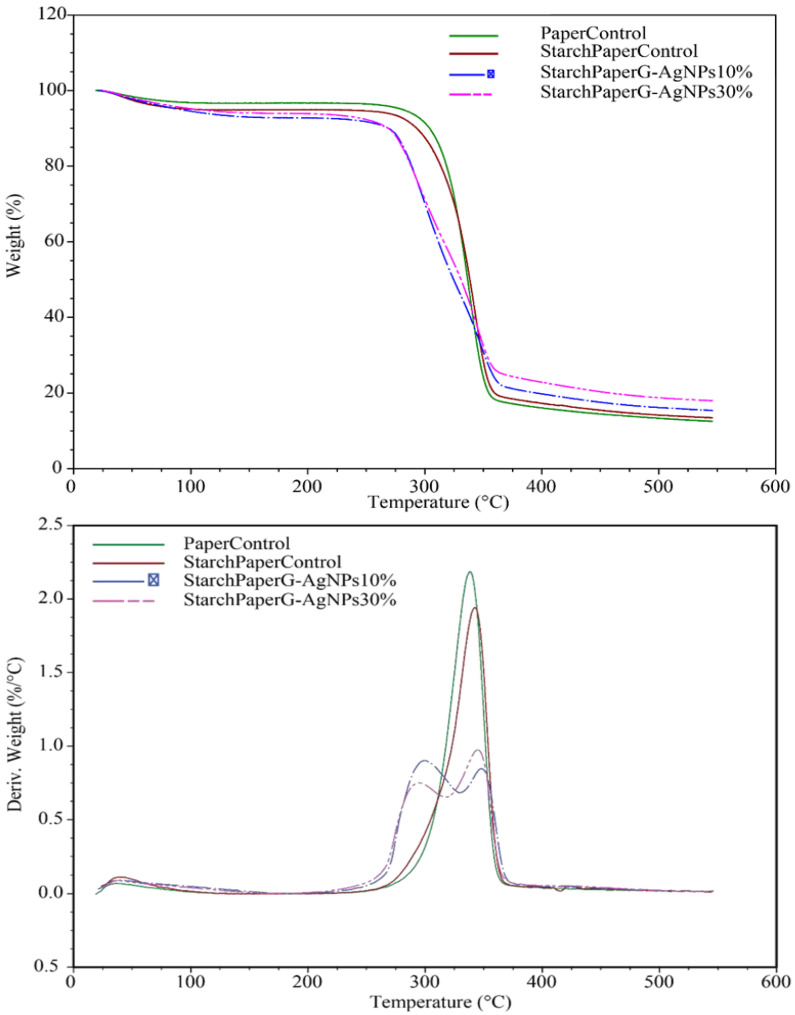
TGA thermograms of starch paper nanocomposites coated with different concentrations of G-AgNPs.

**Figure 8 polymers-16-00941-f008:**
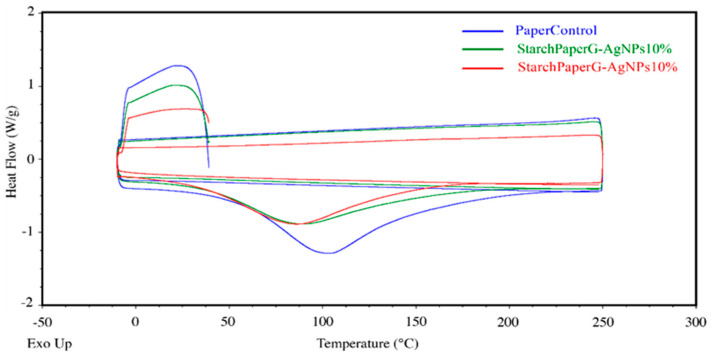
DSC thermograms.

**Figure 9 polymers-16-00941-f009:**
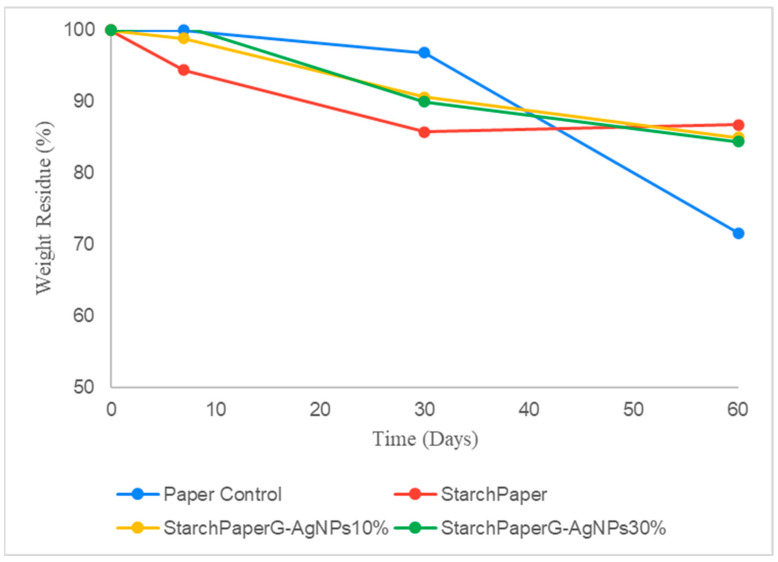
Percentage of weight residue during the bio-disintegration in simulated seawater of functionalised and control papers.

**Figure 10 polymers-16-00941-f010:**
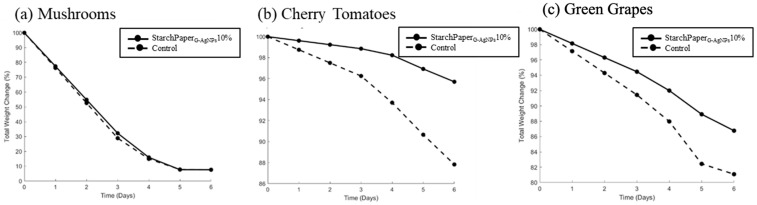
Total weight change per day (%) for (**a**) mushrooms, (**b**) cherry tomatoes, and (**c**) green grapes packed in control (plain paper) and StarchPaperG-AgNPs10% coated paper packaging during storage for 6 days.

**Figure 11 polymers-16-00941-f011:**
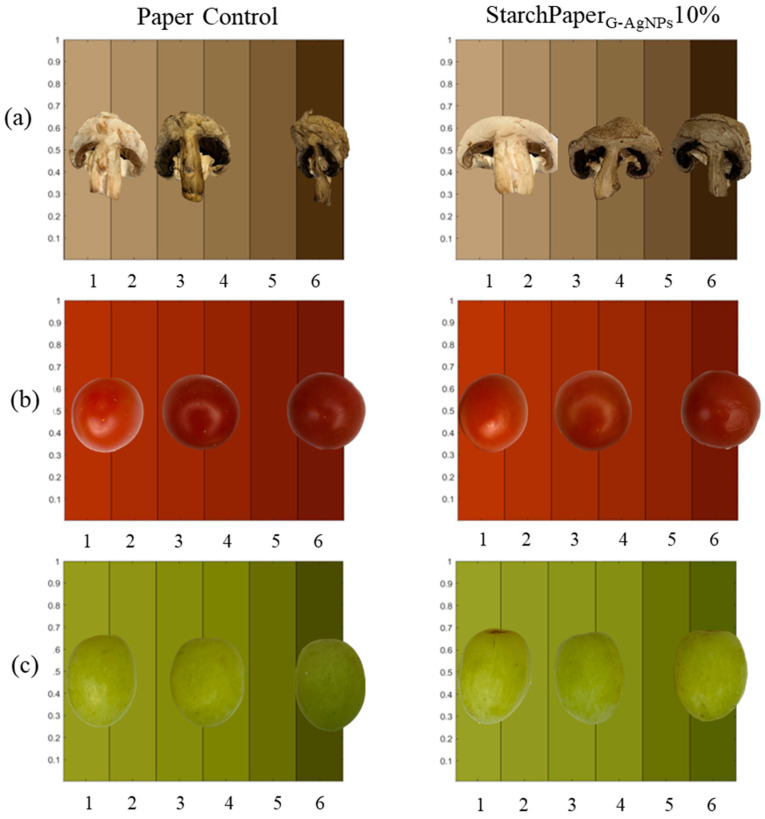
Visual comparison of colour variation from day 0 to day 6 of, (**a**) mushrooms, (**b**) cherry tomatoes, and (**c**) green grapes, packaged in control paper and StarchPaper_G-AgNPs_10%-coated paper packaging. Food pictures at days 1, 3, and 6 of trial.

**Figure 12 polymers-16-00941-f012:**
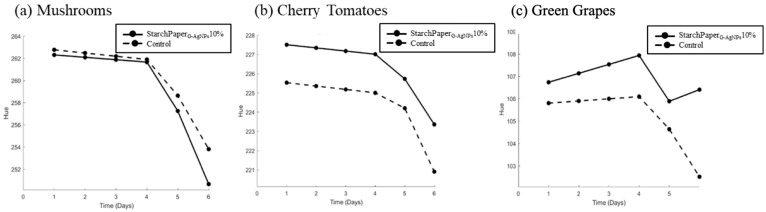
Colour hue angle of (**a**) mushroom, (**b**) cherry tomatoes, and (**c**) green grapes packaged in control paper and StarchPaper_G-AgNPs_10%-coated paper packaging from day 0 to day 6.

**Table 2 polymers-16-00941-t002:** Results of WVTR and WVP of tested paper-based packaging.

Samples	WVTR (g/m^2^/24 h)	WVP (×10^−6^ g/s·m·Pa)	WVP (×10^−6^ g/s·m·Pa) Data Literature [44,45]
PaperControl	135.3 ± 7.2	2.22 ± 0.12	
StarchPaper	127.9 ± 8.2	2.17 ± 0.08	2.60 *
StarchPaper_G-AgNPs_5%	122.6 ± 6.4	2.12 ± 0.03	2.45 **
StarchPaper_G-AgNPs_10%	123.8 ± 5.5	2.15 ± 0.05	
StarchPaper_G-AgNPs_20%	128.4 ± 6.8	2.16 ± 0.10	
StarchPaper_G-AgNPs_30%	117.9 ± 4.8	2.03 ± 0.06	

* Data reported in the literature for starch surface functionalised materials; ** data reported in the literature for starch-AgNPs surface functionalised materials; results obtained with a minimum of three repetitions per sample (*p* value < 0.02).

**Table 3 polymers-16-00941-t003:** Results of WCA for paper samples and drop macroscopical aspect.

Samples	WCA	Water Drop Aspect
PaperControl	82.3° ± 5.3°	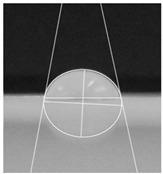
StarchPaper	86.9° ± 2.5°	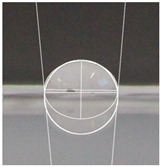
StarchPaper_G-AgNPs_5%	97.1° ± 5.0°	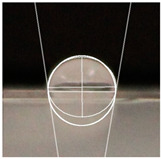
StarchPaper_G-AgNPs_10%	103.2° ± 5.1°	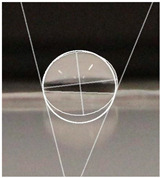
StarchPaper_G-AgNPs_20%	103.9° ± 2.4°	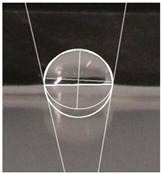
StarchPaper_G-AgNPs_30%	110.0° ± 3.5°	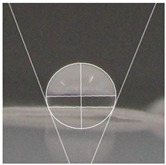

Results obtained with a minimum of 10 repetitions per sample (*p* value < 0.05).

**Table 4 polymers-16-00941-t004:** Results of paper starch materials thermal properties.

Sample	T Peak Max (°C)	Weight Residue (%)
PaperControl	338.4 ± 2.1	12.5 ± 0.9
StarchPaper	339.6 ± 1.6	13.4 ± 1.0
StarchPaper_G-AgNPs_10%	300.2 ± 1.1	15.4 ± 0.7
StarchPaper_G-AgNPs_30%	296.14 ± 1.0	18.0 ± 1.0

**Table 5 polymers-16-00941-t005:** DSC thermal properties.

Sample	Melting Temperature (°C)
PaperControl	103.4 ± 2.8
StarchPaper_G-AgNPs_10%	89 ± 1.3

**Table 6 polymers-16-00941-t006:** Results of inhibition zones of functionalised papers with different AgNPs concentration.

Sample	Zone of Inhibition after 24 h (cm)	Zone of Inhibition after 48 h (cm)	Picture after 24 h
StarchPaper	No inhibition	No inhibition	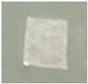
StarchPaper_G-AgNPs_5%	No inhibition	0.105 ± 0.012	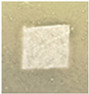
StarchPaper_G-AgNPs_10%	0.114 ± 0.031	0.171 ± 0.032	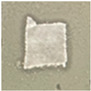
StarchPaper_G-AgNPs_20%	0.141 ± 0.018	0.152 ± 0.022	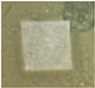
StarchPaper_G-AgNPs_30%	0.158 ± 0.033	0.207 ± 0.044	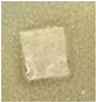

**Table 7 polymers-16-00941-t007:** Pictures of the functionalised papers at different time points of soil bio-disintegration.

Sample	7 Days	30 Days	60 Days
Paper Control	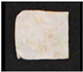	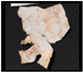	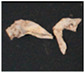
StarchPaper	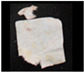	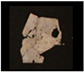	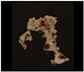
StarchPaper_G-AgNPs_10%	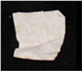	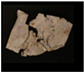	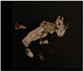
StarchPaper_G-AgNPs_30%	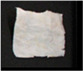	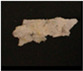	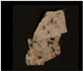

**Table 8 polymers-16-00941-t008:** Starch G-AgNP solutions composition tested.

Sample	G-AgNPs Added Volume (mL)	G-AgNPs Solution Concentration (ppm)	G-AgNPs/Starch wt/wt (%)
StarchPaperControl	0	/	/
StarchPaper_G-AgNPs_5%	2.5	2.12	0.005
StarchPaper_G-AgNPs_10%	5	4.25	0.011
StarchPaper_G-AgNPs_20%	10	8.5	0.021
StarchPaper_G-AgNPs_30%	15	12.75	0.032

## Data Availability

Data will be made available on request.

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
