# Peer review of "Advancing Food Preservation: Sustainable Green-AgNPs Bionanocomposites in Paper-Starch Flexible Packaging for Prolonged Shelf Life"

_polymers, 2024, doi:10.3390/polym16070941_

Round 1

Reviewer 1 Report

Comments and Suggestions for Authors

1. Materials and methods of all experiments are needed to be described. The number of replications and details of the instruments should be added. Statistical analysis should be added. This section should come before the results.

2. Results : Add the SD values for all experiments e.g. Table 4 and 5 along with the statistical analysis

3. Recheck the spelling e.g. L 327 " his...", 

4 Recheck the scale of Fig. 10

5. Recheck all the citation once again

L345-348 is confusing whether it is the present discussion or from previous work or the general statement.

L211 how mechanical properties explain the durability?

L235-236 why the moisture barrier relate to potential contaminants.

Table 3 recheck the angle for starch paper. It said 90 degree but the line seems to be different

Fig. 7 ; It is difficult to see the color

Comments on the Quality of English Language

It can be improved.

Reviewer 2 Report

Comments and Suggestions for Authors

This study investigates the development of corn-starch-based coating materials reinforced with green silver nanoparticles (G-AgNPs) for food packaging applications. Mechanical properties, water vapor permeability (WVP), antibacterial activity against E. coli, and environmental degradation were evaluated. Food trial analysis was also conducted to assess the effectiveness of the packaging in preserving cherry tomatoes and green grapes.

Questions:

Do not use many times words: in our study.

The aim at the end of the Introduction part is too long, it should be concise and without the description of the material and methods.

Figures with SEM should be of better quality.

How does the incorporation of green silver nanoparticles (G-AgNPs) enhance the mechanical properties of corn-starch-based packaging materials?

What specific characteristics of G-AgNPs contribute to their antibacterial activity against E. coli?

Can you elaborate on the mechanisms through which the hydrophobic layer, formed by G-AgNPs, reduces water vapor permeability in the packaging material?

In what ways do the results of this study align with the United Nations Sustainable Development Goals (SDGs) related to reducing food waste and promoting sustainability?

Are there any potential concerns or limitations associated with the use of G-AgNPs in food packaging materials, particularly in terms of environmental impact or safety?

How do the findings of this study contribute to the ongoing efforts to develop environmentally friendly packaging materials for food preservation?

Could the methodology used in this study be adapted or modified for other types of food products or packaging materials? If so, what modifications might be necessary?

Reviewer 3 Report

Comments and Suggestions for Authors

See the attachment

Round 2

Reviewer 1 Report

Comments and Suggestions for Authors

The manuscript has been improved.

Comments on the Quality of English Language

Good